# FAST AGNOSTIC LEARNERS IN THE PLANE

## ABSTRACT

We investigate the computational efficiency of agnostic learning for several fundamental geometric concept classes in the plane. While the sample complexity of agnostic learning is well understood, its time complexity has received much less attention. We study the class of triangles and, more generally, the class of convex polygons with $k$ vertices for small $k$, as well as the class of convex sets in a square. We present a proper agnostic learner for the class of triangles that has optimal sample complexity and runs in time $\tilde{O}(\epsilon^{-6})$, improving on the algorithm of Dobkin and Gunopulos (COLT '95) that runs in time $\tilde{O}(\epsilon^{-10})$. For 4-gons and 5-gons, we improve the running time from $O(\epsilon^{-12})$, achieved by Fischer and Kwek (eCOLT '96), to $\tilde{O}(\epsilon^{-8})$ and $\tilde{O}(\epsilon^{-10})$, respectively.

We also design a proper agnostic learner for convex sets under the uniform distribution over a square with running time $\tilde{O}(\epsilon^{-5})$, improving on the previous $\tilde{O}(\epsilon^{-8})$ bound at the cost of slightly higher sample complexity. Notably, agnostic learning of convex sets in $[0,1]^2$ under general distributions is impossible because this concept class has infinite VC-dimension. Our agnostic learners use data structures and algorithms from computational geometry and their analysis relies on tools from geometry and probabilistic combinatorics. Because our learners are proper, they yield tolerant property testers with matching running times. Our results raise a fundamental question of whether a gap between sample and time complexity is inherent for agnostic learning of these and other natural concept classes.

## 1 INTRODUCTION

The agnostic learning framework, introduced by Haussler (1992) and Kearns et al. (1994), elegantly models learning from noisy data. It has had a tremendous impact, driving advances in both theory and practice. Theoretically, it has deepened our understanding of learning theory, data compression, computational complexity, property testing, and more. Practically, it underpins robust algorithms for image recognition, signal processing, adversarial learning, and beyond. While the sample complexity of agnostic learning is well understood, its time complexity remains underexplored.

Halfspaces are among the most fundamental concept classes in theory and practice, but their limited expressiveness has led to interest in richer models. For instance, Kantchelian et al. (2014) showed that convex polytope classifiers outperform hyperplanes in both speed and accuracy. However, expressive classes in high dimensions face a major obstacle: learning is computationally hard, even for halfspaces. While PAC learning of $d$-dimensional halfspaces is efficient in the realizable setting, it is NP-hard agnostically (Guruswami & Raghavendra (2009); Feldman et al. (2009; 2012)). In contrast, in 2D, halfplanes admit efficient agnostic learners with optimal sample and time complexity $\widetilde{O}(\frac{1}{\epsilon^2}\log\frac{1}{\delta})$ (Matheny & Phillips (2021)). This raises a natural question: do other expressive *2D* classes – polygons and convex sets – admit comparably efficient agnostic learners? More broadly, this points to a theme: *low-dimensional geometric classes may offer algorithmic advantages that are not yet well understood.*

We investigate the computational efficiency of agnostic learners for several fundamental geometric concept classes in the plane, namely, of triangles, convex $k$-gons, and convex sets in a square. These classes arise naturally in computer vision, image analysis, and shape recognition, where data is often low-dimensional and spatially structured. Moreover, they serve as canonical examples in geometric learning theory, providing a clean yet expressive setting for studying the trade-offs between sample complexity and computational efficiency. We focus on *proper* agnostic learners—those that output hypotheses from the concept class being learned—as this is crucial for our application to tolerant property testing. All learners we discuss, including those from prior work, are proper.

The class of $k$-gons (that is, convex polygons with $k$ vertices) has VC-dimension $2k + 1$ (Dobkin & Gunopulos (1995)), and thus, by standard VC-dimension bounds, the sample complexity of agnostically PAC-learning this class is $s = \Theta(\frac{1}{\varepsilon^2}(k + \ln \frac{1}{\delta}))$. The running time of agnostically learning $k$-gons has been investigated by Fischer (1995), Dobkin & Gunopulos (1995), and Fischer & Kwek (1996). Specifically, Dobkin & Gunopulos (1995) designed a (proper) agnostic learner for $k$-gons with running time $O(s^{2k-1} \log s)$, where $s$ is the size of the sample. For triangles (the case $k = 3$), the running time has $\widetilde{O}(\frac{1}{\varepsilon^{10}})$ dependence on the loss parameter $\varepsilon$, which, to our knowledge, is currently the best bound for this case. Prior to our work, the best bound on the running time for larger $k$ was $O(ks^6)$ in terms of the sample size $s$, by Fischer & Kwek (1996). Thus, the dependence on $\varepsilon$ in the running time was $O(\frac{1}{\varepsilon^{12}})$ for 4-gons and 5-gons. Notably, agnostic learners for these basic concept classes suffered from high running times. We give a proper agnostic PAC learner for $k$-gons for constant $k$ with *optimal sample complexity* and running time $\tilde{O}(\frac{1}{\varepsilon^{2k}} \log \frac{1}{\delta} + \frac{1}{\varepsilon^2} \log^2 \frac{1}{\delta})$. This improves the dependence on $\varepsilon$ to $\tilde{O}(\frac{1}{\varepsilon^6})$ for triangles, $\tilde{O}(\frac{1}{\varepsilon^8})$ for 4-gons, and $\tilde{O}(\frac{1}{\varepsilon^{10}})$ for 5-gons, making progress on long-standing open questions.

We also study agnostic learning of general convex sets in the plane under the uniform distribution over a unit square. This concept class, denoted $\mathcal{C}_{\text{conv}}$, has infinite VC-dimension and thus is not PAC learnable under general distributions. Motivated by property testing of images, Berman et al. (2022) gave a (proper) agnostic learner for $\mathcal{C}_{\text{conv}}$ under the uniform distribution over a square. Their learner has sample complexity $O(\frac{1}{\varepsilon^2} \log \frac{1}{\varepsilon\delta})$ and time complexity $\widetilde{O}(\frac{1}{\varepsilon^8} + \frac{1}{\varepsilon^7} \log^2 \frac{1}{\delta})$. Moreover, they showed that $\Omega(\frac{1}{\varepsilon^2})$ samples are required for this task. We design a (proper) agnostic learner for this task with a significantly improved running time at the expense of a small overhead in sample complexity. Our learner takes a sample of size $\widetilde{O}(\frac{1}{\varepsilon^{2.5}} \log \frac{1}{\delta})$ and runs in time $\widetilde{O}(\frac{1}{\varepsilon^5} \log \frac{1}{\delta} + \frac{1}{\varepsilon^2} \log^2 \frac{1}{\delta})$. It leaves an intriguing question of whether there is an inherent tradeoff between the sample complexity and the running time of agnostic learning of this class. Our results on proper agnostic learners are summarized and compared to previous work in Table 1.

| | | **Triangles** | **4-gons** | **5-gons** | $\mathcal{C}_{\text{conv}}$ |
|---|---|---|---|---|---|
| **Prior work** | Sample complexity | \multicolumn{3}{c\|}{$s = \Theta(\epsilon^{-2})$ (by VC-dimension)} | $\tilde{O}(\epsilon^{-2})$ [BMR22] |
| | Runtime | $O(\epsilon^{-10})$ [DG95] | \multicolumn{2}{c\|}{$O(\epsilon^{-12})$ [FK96]} | $\tilde{O}(\epsilon^{-8})$ |
| **Our results** | Sample complexity | \multicolumn{3}{c\|}{$s = \Theta(\epsilon^{-2})$} | $\tilde{O}(\epsilon^{-2.5})$ |
| | Runtime | $\tilde{O}(\epsilon^{-6})$ | $\tilde{O}(\epsilon^{-8})$ | $\tilde{O}(\epsilon^{-10})$ | $\tilde{O}(\epsilon^{-5})$ |

Table 1: Comparison of complexity of proper agnostic learners for concept classes we study, stated for constant $\delta$. The class $\mathcal{C}_{\text{conv}}$ is the class of convex sets under the uniform distribution in $[0, 1]^2$.

Following the established connection between PAC learning and property testing (see Goldreich et al. (1998); Berman et al. (2022)), our proper agnostic learners yield tolerant property testers for the geometric classes we study, with the same running times. This is discussed in detail in Appendix A. The fact that our learners are proper is crucial for this application.

## 1.1 OUR TECHNIQUES

**Two-sample framework.** Both of our learners follow the same high-level template: they draw a small sample $N$ to generate a compact *reference family* of candidate hypotheses, and a larger sample $S$ to evaluate their empirical risk. Set $N$ could be a subset of $S$ (as in our result on $k$-gons) or an independent sample (as in our result on general convex sets). The key is to prove that this restricted family still contains a near-optimal hypothesis, so that empirical risk minimization over it gives strong guarantees. This design improves running time by decoupling the (expensive) hypothesis construction from the (cheaper) risk evaluation.

**Agnostic learner for $k$-gons.** Our learner generates all halfplanes induced by pairs of points in sample $N$ and intersect $k$ of them to form candidate $k$-gons. To evaluate each candidate, the learner triangulates it and applies the triangle range-counting data structure of Goswami et al. (2004). Using this building block, the learner efficiently computes the asymmetric discrepancy (see Definition B.6) with respect to $S$ of every reference triangle, which later yields the asymmetric discrepancy of the reference $k$-gons. The analysis relies on the elegant reference halfplane construction of Matheny & Phillips (2018b): we lift their $\varepsilon$-net for halfplanes (via a union bound over the $k$ sides) to show that the candidate family induced by $N$ forms an $\varepsilon$-reference set for $k$-gons. Thus, even though the search space is drastically smaller than the set of all $k$-gons, it suffices to approximate the best one, leading to optimal sample complexity and improved dependence on $\varepsilon$.

**Agnostic learner for convex sets.** Our learner for convex sets uses the smaller sample $N$ to implicitly construct *islands*. An *island* induced by $N$ is a set formed by intersecting $N$ with a convex set. Our learner then evaluates the empirical risk of islands with respect to the larger sample $S$. As in the case of $k$-gons, the learner uses the algorithm of Goswami et al. (2004) to construct a data structure that quickly computes asymmetric discrepancy on each queried triangle. Since the islands considered are induced by a small set, our learner can quickly choose the island with smallest asymmetric discrepancy by running the algorithm of Bautista-Santiago et al. (2011). Given access to triangle queries, their algorithm uses dynamic programming to find an island with maximum asymmetric discrepancy, implying minimum empirical risk. In contrast to our simple randomized construction of islands, the set of reference polygons used in prior work by Berman et al. (2022) is quite complicated: at a high level, it is obtained (deterministically) by taking axis-parallel reference rectangles and iteratively chipping away triangles from the corners of current polygons.

To analyze our learner for convex sets, we show that any convex set (including the optimal set) is closely approximated by the largest island that fits inside it. The behavior of the "missing area" between a convex set and the convex hull of a uniform sample of a given size inside the set has been extensively studied (see, e.g., Har-Peled (2011) and the survey in Bárány (2007)). We utilize the concentration result of Brunel (2020) that demonstrates that if $\ell$ points are sampled from a convex set then the fraction of the missing area is concentrated around $\Theta(\ell^{-2/3})$. It allows us to show that, with high probability, there is an island induced by the sample $N$ that closely approximates the optimal convex set.

In the second part of the analysis, we bound the size of the sample $S$ required to estimate the empirical risk of the islands induced by $N$. Since the class of convex sets has infinite VC-dimension, standard uniform convergence results do not apply. Instead, we first use the concentration bounds of Valtr (1994; 1995) regarding the maximum convex set of points in a uniform sample from $[0,1]^2$ to show that all islands are likely to have $O(\sqrt[3]{|N|})$ vertices in their convex hulls. Then we apply uniform convergence to all convex sets on $O(\sqrt[3]{|N|})$ vertices to get a bound on the size of a representative sample.

## 1.2 OPEN PROBLEMS AND FUTURE DIRECTOINS

Our agnostic learners have better runtime than previously know, in some cases improving decades-old classical algorithms. However, their running time is still larger than their sample complexity. It is an interesting (and difficult) open problem to either improve the running time or to justify this discrepancy by proving

computational hardness, for example, by using the tools from fine-grained complexity. For the case of agnostic learners of $k$-gons, we were able to improve the running time for triangles, 4-gons, and 5-gons. For larger constant $k$, the best known running time is $\widetilde{O}(\frac{1}{\varepsilon^{12}})$, and it is open whether it can be improved. For the case of agnostic learning of convex sets, our algorithm is faster than that of Berman et al. (2022), but has slightly worse sample complexity. It is open whether it is possible to get a fast algorithm with optimal sample complexity.

**Future directions.** Although our results focus on two dimensions—where efficient algorithms are more within reach—gaining a precise understanding of these settings is a crucial step toward addressing higher-dimensional cases. Another natural direction is to relax the requirement of properness: while proper learners are particularly valuable due to their connections to property testing and related applications, it remains an intriguing open question whether improper learners can be more efficient.

### 1.3 RELATED WORK

Agnostic PAC learning, introduced by Haussler (1992) and Kearns et al. (1994), has played a central role in learning theory. For halfspaces, PAC learning is efficient in the realizable case, but agnostic learning is NP-hard in high dimensions [Guruswami & Raghavendra (2009); Feldman et al. (2009; 2012)]. In contrast, 2D halfplanes admit efficient agnostic learners with optimal sample and time complexity [Matheny & Phillips (2021)]. Intersections of halfspaces have also been studied, with hardness results showing that efficient agnostic learning is unlikely in general [Giannopoulos et al. (2012); Daniely et al. (2014)]. On the algorithmic side, Dobkin & Gunopulos (1995) and Fischer & Kwek (1996) designed early learners for polygons, while Berman et al. (2022) studied convex sets under the uniform distribution, proving near-optimal sample bounds but with high running time. Works on related problems in agnostic learning and computational geometry are too numerous to list here. We mention a couple. Kwek & Pitt (1996) presented a PAC learning algorithm with *membership queries* for a class of intersections of halfspaces in $d$-dimensional space. Eppstein et al. (1992) showed how to find, given a set $P$ of $n$ points with weights, a maximum-weight $k$-gon with vertices in $P$ in time $O(kn^3)$.

### 1.4 PRELIMINARIES

Due to space limitations, we defer the preliminaries to Appendix B.

## 2 AGNOSTIC LEARNER AND ERM FOR $k$-GONS

This section proves our result on agnostic learning of $k$-gons. The following theorem, stated for all integers $k \geq 3$, improves upon previous bounds for triangles, quadrilaterals, and pentagons.

**Theorem 2.1.** $\forall \varepsilon, \delta \in (0, 1)$ *and constant integer $k \geq 3$, the class of $k$-gons over $\mathbb{R}^2$ is properly agnostically PAC learnable with* $O(\frac{1}{\varepsilon^2} \log \frac{1}{\delta})$ *samples and in* $O(\frac{1}{\varepsilon^{2k}} (\log \frac{1}{\varepsilon}) \log \frac{1}{\delta} + \frac{1}{\varepsilon^4} \log^2 \frac{1}{\delta})$ *time.*

Let $\mathcal{C}_k$ denote the class of $k$-gons over $\mathbb{R}^2$ (as in Definition B.3). Our $k$-gon learner first obtains a sample $S$ of $s$ examples from distribution $\mathcal{D}$, where $s$ is large enough to get uniform convergence for the class $\mathcal{C}_k$ with loss parameters $\frac{\varepsilon}{3}$ and constant failure probability. The class of $k$-gons has VC-dimension $2k + 1$ (Dobkin & Gunopulos (1995)). By standard VC-dimension arguments ((Shalev-Shwartz & Ben-David, 2014, Theorem 6.8)), a sample of size $s = O(\frac{1}{\varepsilon^2} \log \frac{1}{\delta})$ is sufficient when $k$ is constant and, moreover, an algorithm that finds an empirical risk minimizer on such a sample is an agnostic PAC learner (see discussion in Appendix B.3). Our algorithm finds a hypothesis that approximately minimizes the risk. Its performance is summarized in Theorem 2.2. Later (in Appendix E), we use our ERM minimizer on a sample of size $s$ and then amplify the success probability of the resulting learner to complete the proof of Theorem 2.1.

**Theorem 2.2.** *For all $\varepsilon \in (0,1)$ and fixed $k \geq 3$, there is an algorithm (specifically, Algorithm 1) that finds a hypothesis with empirical risk at most $OPT + \varepsilon$ from the class $\mathcal{C}_k$ (of $k$-gons over $\mathbb{R}^2$) on a set $S$ of examples in time $O(\frac{1}{\varepsilon^{2k}} \log |S| + |S|^2)$ with success probability at least $\frac{2}{3}$, where $OPT$ is the smallest empirical risk of a hypothesis in $\mathcal{C}_k$ on the set $S$.*

To compute the empirical risk of a $k$-gon, we use the triangle range–counting data structure of Goswami et al. (2004). Recall that ERM is equivalent to maximizing asymmetric discrepancy (Def. B.6, Claim B.7). The algorithm of Goswami et al. pre-processes a point set in the plane so that, given a query triangle $T$, it quickly returns the number of points in $T$. We build two such structures, one for positive and one for negative examples, allowing us to compute the discrepancy of any query triangle with two fast queries.

**Theorem 2.3** (Corollary of Theorem 2, Goswami et al. (2004)). *There exists an algorithm $\mathcal{B}_{pre}$ that, for any set $S \subset \mathbb{R}^2 \times \{0, 1\}$, builds a data structure DS of size $O(|S|^2)$ in time $O(|S|^2)$ for computing asymmetric discrepancy of triangles w.r.t. $S$. There also exits a query algorithm $\mathcal{B}_{query}$ that, given the data structure DS and a (geometric) triangle $T$, returns in $O(\log |S|)$ time the asymmetric discrepancy of $T$ on $S$.*

Our ERM constructs a reference set of $k$-gons. The notion of a reference set is defined next.

**Definition 2.4** (An $\varepsilon$-reference set). For functions $h, r : \mathcal{X} \to \{0, 1\}$, let $h \oplus r$ denote the XOR function, i.e., $(h \oplus r)(x) = h(x) + r(x) \mod 2$. Let $\mathcal{C}$ be a class of concepts from $\mathcal{X}$ to $\{0, 1\}$ and $\mathcal{R}, H \subset \mathcal{C}$. The set $\mathcal{R}$ is an *$\varepsilon$-reference set* for a concept class $\mathcal{C}$ w.r.t. a set $S$ of examples and a set $H$ if $\forall h \in H$, there exists a hypothesis $r \in \mathcal{R}$ such that $|\{(x, y) \in S : (h \oplus r)(x) = 1\}| \leq \varepsilon |S|$.

In other words, for every $h \in H$, there exists some reference hypothesis $r \in \mathcal{R}$ that does not differ too much from $h$ on the points in $S$. For brevity, when $H = \{h\}$, we say "a reference set for $h$" instead of "for $H$". Next we draw a random sample $N$ to specify the specific set of $k$-gons that we use as a reference set.

**Definition 2.5** (Induced halfplanes, reference $k$-gons). A halfplane $h \in \mathcal{H}^2$ is *induced* by a set $N \subset \mathbb{R}^2$ if $h$ is defined by a line that passes through two points in $N$. Let $\mathcal{I}_N$ denote the set of all halfplanes induced by $N$. A $k$-gon $t$ in $\mathcal{C}_k$ is a *reference $k$-gon defined by a set* $N \subset \mathbb{R}^2$ if $t$ is formed by the intersection of $k$ induced halfplanes. Let $\mathcal{R}_N$ be the set of all reference $k$-gons defined by $N$.

Algorithm 1 constructs a data structure DS for computing asymmetric discrepancy of triangles on sample points. For every $k$-gon $P \in \mathcal{R}_N$, it computes its asymmetric discrepancy $disc_S(P)$ by triangulating $P$ and querying DS on each triangle $T$ in the triangulation and summing up the results. The algorithm returns the indicator function for the $k$-gon with the largest asymmetric discrepancy.

---

**Algorithm 1:** Algorithm for Approximating ERM for $k$-gons

**input** : loss parameter $\varepsilon \in (0, 1)$; a set $S \subseteq \mathbb{R}^2 \times \{0, 1\}$ of examples.

1 Run algorithm $\mathcal{B}_{pre}$ from Theorem 2.3 on the set of examples $S$ to construct a data structure DS for computing asymmetric discrepancy of triangles w.r.t. $S$.

2 Sample a set $N$ of $n = \frac{ck \log k}{\epsilon}$ points from $\{x \in \mathbb{R}^2 : (x, y) \in S\}$ uniformly and independently with replacement, where $c$ is a large enough constant (dictated by Lemma 2.8).

3 Compute the reference set $\mathcal{R}_N$ of $k$-gons (see Definition 2.5).

4 For each $k$-gon $f_P$ in $\mathcal{R}_N$, compute the triangulation of $P$ into $k - 2$ triangles $T_1, \ldots, T_{k-2}$. Use algorithm $\mathcal{B}_{query}$ from Theorem 2.3 to query the data structure DS on each triangle $T_i$ for $i \in [k - 2]$ to get asymmetric discrepancy $disc_S(T_i)$. Compute $disc_S(P) = \sum_{i \in [k-2]} disc_S(T_i)$.
   `\\Counts can be easily adjusted for boundary points.`

5 Return the $k$-gon $f_{\tilde{P}}$, where $\tilde{P} = \arg\max_{P \in \mathcal{R}_N} disc_S(P)$.

---

## 2.1 ANALYSIS OF THE ERM FOR $k$-GONS

To analyze correctness of Algorithm 1, we show that $\mathcal{R}_N$ is likely to be a good reference set for $\mathcal{C}_k$ w.r.t. the input set $S$ of examples and the ERM $k$-gon that labels them optimally. We start by proving that any good reference set for halfplanes yields a good reference set for $k$-gons when used in our construction.

**Lemma 2.6.** *Fix a loss parameter $\varepsilon \in (0, 1)$, an integer $k \geq 3$, a set of examples $S$, and a $k$-gon $t \in \mathcal{C}_k$. Let $t$ be the intersection of $k$ halfplanes $h_1, \ldots, h_k \in \mathcal{H}^2$. Let $R \subset \mathcal{H}^2$ be an $\frac{\varepsilon}{k}$-reference set for $\mathcal{H}^2$ w.r.t. $S$ and $\{h_1, \ldots, h_k\}$. Let $K_R$ be the set of all $k$-gons formed by an intersection of $k$ halfplanes in $R$. Then $K_R$ is an $\varepsilon$-reference set for $\mathcal{C}_k$ w.r.t. $S$ and the $k$-gon $t$.*

*Proof.* Since $R$ is an $\frac{\varepsilon}{k}$-reference set for $\mathcal{H}^2$ w.r.t. $S$ and $\{h_1, \ldots, h_k\}$, there exist halfplanes $h'_1, \ldots, h'_k \in R$ such that $|\{(x, y) \in S : (h_i \oplus h'_i)(x) = 1\}| \leq \frac{\varepsilon}{k}|S|$ for all $i \in [k]$. Let $t'$ be the $k$-gon formed by the intersection of halfplanes $h'_1, \ldots, h'_k$. Then $t' \in K_R$. The $k$-gons $t$ and $t'$ can differ only on points on which at least one of the corresponding pairs of halfplanes differs:

$$|\{(x, y) \in S : (t \oplus t')(x) = 1\}| \leq \sum_{i \in [k]} |\{(x, y) \in S : (h_i \oplus h'_i)(x) = 1\}| \leq k \frac{\varepsilon}{k}|S| = \varepsilon|S|.$$

Thus, for the $k$-gon $t$, there is a nearby (w.r.t. $S$) reference $k$-gon $t' \in K_R$, as required. $\qquad\square$

To obtain a good reference set for halfplanes, we use following result.

**Lemma 2.7** (Matheny & Phillips (2018a)). *Fix a concept $h \in \mathcal{H}^2$ and a set $S$ of examples from $\mathbb{R}^2 \times \{0, 1\}$. If a set $N$ of size $\frac{4}{\varepsilon} \ln \frac{2}{\delta_0}$ is sampled uniformly and independently with replacement from $\{x : (x, y) \in S\}$ then the induced set $\mathcal{I}_N$ is an $\varepsilon$-reference set for $\mathcal{H}^2$ w.r.t. $S$ and $h$ with probability at least $1 - \delta_0$.*

Using this lemma, we obtain the following guarantee for the reference set $\mathcal{R}_N$.

**Lemma 2.8.** *Fix a concept $t \in \mathcal{C}_k$ and a set $S$ of examples from $\mathbb{R}^2 \times \{0, 1\}$. If a set $N$ of size $c\frac{k}{\varepsilon} \log k$, where $c$ is a sufficiently large constant, is sampled uniformly and independently with replacement from $\{x : (x, y) \in S\}$ then the set $\mathcal{R}_N$ is an $\varepsilon$-reference set for $\mathcal{C}_k$ w.r.t. $S$ and $t$ with constant probability.*

*Proof.* Let $h_1, \ldots, h_k \in \mathcal{H}^2$ be the halfplanes such that the $k$-gon $t$ is formed by their intersection. By Lemma 2.7, for $N$ of size $\frac{4k}{\varepsilon} \ln \frac{2k}{\delta_0}$, the set $\mathcal{I}_N$ fails to be an $\frac{\varepsilon}{k}$-reference set for $\mathcal{H}^2$ w.r.t. $S$ and one specific $h_i$, where $i \in [k]$, with probability at most $\frac{\delta_0}{k}$. By a union bound, we get that $\mathcal{I}_N$ fails to be an $\frac{\varepsilon}{k}$-reference set for $\mathcal{H}^2$ w.r.t. $S$ and $\{h_1, \ldots, h_k\}$ with probability at most $\delta_0$. By Lemma 2.6, if no failure events occur, then $\mathcal{R}_N$ is an $\varepsilon$-reference set for $\mathcal{C}_k$ w.r.t. $S$ and $k$-gon $t$. We set $\delta_0 = \frac{1}{3}$ to obtain the desired statement. $\qquad\square$

*Proof of Theorem 2.2.* First, we analyze correctness of Algorithm 1. Let $t^* \in \mathcal{C}_k$ be a $k$-gon which achieves the minimum error on $S$. Namely, $err_S(t^*) \leq err_S(t)$ for all $t \in \mathcal{C}_k$. By Lemma 2.8, the set $\mathcal{R}_N$ constructed by Algorithm 1 is *good*, i.e., an $\varepsilon$-reference set for $\mathcal{C}_k$ w.r.t. $S$ and $t^*$, with constant probability. Now suppose $\mathcal{R}_N$ is good. We analyze the error of the hypothesis $f_{\tilde{P}}$ returned by Algorithm 1. Let $OPT = err_S(t^*)$. Since $\mathcal{R}_N$ is good, there is a reference $k$-gon $r \in \mathcal{R}_N$ such that $|err_S(t^*) - err_S(r)| \leq \varepsilon$ and, consequently, $err_S(r) \leq err_S(t^*) + \varepsilon = OPT + \varepsilon$. Since Algorithm 1 outputs a reference $k$-gon with the largest asymmetric discrepancy, and thus (by Claim B.7) with the smallest empirical risk, we get $err_S(f_{\tilde{P}}) \leq err_S(r)$. Combining this with the above implies $err_S(f_{\tilde{P}}) \leq err_S(r) \leq OPT + \varepsilon$. Therefore, $err_S(f_{\tilde{P}}) \leq OPT + \varepsilon$ with constant probability.

**Running time:** The most time consuming steps of Algorithm 1 are Steps 1 and 4. By Theorem 2.3, the running time of the preprocessing step is $O(|S|^2)$. Each query to the data structure DS takes time $O(\log(|S|))$. In Step 4, the algorithm queries DS on each element of $\mathcal{R}_N$. There are at most $|N|^{2k}$ reference $k$-gons in the

set $\mathcal{R}_N$. For each reference $k$-gon, the algorithm computes its triangulation in time $O(k)$ and queries DS on each of the $k$ triangles from the triangulation. Hence, Step 4 takes $O\left(\left(\frac{k \log k}{\varepsilon}\right)^{2k} \cdot k \log |S|\right) = O\left(\frac{1}{\varepsilon^{2k}} \log |S|\right)$ time for constant $k$. The total running time of Algorithm 1 is thus $O\left(\frac{1}{\varepsilon^{2k}} \cdot \log |S| + |S|^2\right)$. $\qquad\square$

## 3 AGNOSTIC LEARNER FOR CONVEX SETS

We start by stating the guarantees of our learner for convex sets over $[0,1]^2$ (see Definition B.4).

**Theorem 3.1.** *For all $\varepsilon, \delta \in (0,1)$, the class $\mathcal{C}_{conv}$ of convex sets over $\mathcal{X} = [0,1]^2$ is properly agnostically PAC learnable with $O\left(\frac{1}{\varepsilon^{2.5}} \ln \frac{1}{\varepsilon} \cdot \ln \frac{1}{\delta}\right)$ samples and running time $O\left(\frac{1}{\varepsilon^5} \ln^2 \frac{1}{\varepsilon} \cdot \ln \frac{1}{\delta} + \frac{1}{\varepsilon^2} \ln \frac{1}{\varepsilon} \ln^2 \frac{1}{\delta}\right)$ under each distribution $\mathcal{D}$ over $\mathcal{X} \times \{0,1\}$, where the marginal $\mathcal{D}_\mathcal{X}$ is uniform over $\mathcal{X}$.*

We first present and analyze Algorithm 2, our learner for convex sets that has constant failure probability. In Appendix E, we use standard arguments to amplify the success probability of Algorithm 2 to $1 - \delta$ and complete the proof of Theorem 3.1. The guarantees of Algorithm 2 are stated next.

**Theorem 3.2.** *For all $\varepsilon \in (0,1)$, Algorithm 2 takes a sample of size $O\left(\frac{1}{\varepsilon^{2.5}} \ln \frac{1}{\varepsilon}\right)$ from a distribution $\mathcal{D}$ over $\mathcal{X} \times \{0,1\}$, where the marginal distribution $\mathcal{D}_\mathcal{X}$ over $\mathcal{X}$ is uniform, and returns, with probability at least $\frac{2}{3}$, a hypothesis $h$ which is an indicator function for a polygon on $O(\varepsilon^{-0.5})$ vertices and satisfies $err_\mathcal{D}(h) \leq \min_{f \in \mathcal{C}_{com}}\{err_\mathcal{D}(f)\} + \varepsilon$. The running time of Algorithm 2 is $O\left(\frac{1}{\varepsilon^5} \ln^2 \frac{1}{\varepsilon}\right)$.*

Algorithm 2 starts by obtaining a sample $S$ from $\mathcal{D}$ and a net $N$ drawn uniformly at random from $[0,1]^2$. The net $N$ is used to construct reference objects, called *islands*. Intuitively, an island induced by a set $N$ is a subset of $N$ formed by an intersection of $N$ and some convex set. See Figure 1. Our algorithm outputs the indicator function of the island induced by the net $N$ that has the largest asymmetric discrepancy, and thus the smallest empirical risk, w.r.t. the set $S$.

**Definition 3.3** (Island). *An island $I$ induced by a set $N \subseteq \mathbb{R}^2$ is a subset $I \subseteq N$ such that $\text{Hull}(I) \cap N = I$. The polygon $\text{Hull}(I)$ defined by an island $I$ is denoted $P_I$ (recall that $\text{Hull}(I)$ denotes the convex hull of $I$). Let $\mathcal{I}_N$ be the set of all islands induced by $N$ and $\mathcal{T}_N$ be the set of all triangles induced by $N$ (i.e., triangles with vertices in $N$). Let $f_I$ be the indicator function for the polygon $P_I$.*

Our algorithm relies on the OptIslands procedure of Bautista-Santiago et al. (2011), which optimizes any monotone decomposable function $\alpha$ over polygons. Intuitively, such functions decompose into contributions from smaller polygons.[1] See Appendix D for the formal definitions and guarantees about OptIslands. To invoke OptIslands, we define $\alpha(P) = disc_S(P)$ for every convex polygon $P$, where $S$ is a sample drawn from $\mathcal{D}$. Then $\alpha$ is decomposable and monotone, and we can use Theorem 2.3 to build a data structure for computing $\alpha$ for all triangles induced by the set $N$ in time $O(|S|^2 + |N|^3 \log |S|)$. Therefore, OptIslands returns the island of $N$ maximizing asymmetric discrepancy among all islands in $\mathcal{I}_N$.

The analysis of Algorithm 2 is organized into three sections. In Section 3.1, we show that with sufficient probability there is an island that approximates the best convex set well. In Section 3.2, we prove that the empirical risk is accurate for all islands with sufficient probability. Finally, in Section D.4, we put everything together and complete the proof of Theorem 3.2.

### 3.1 A NEARLY OPTIMAL ISLAND

In this section, we show that for a sample set $N$ of size $O\left(\frac{1}{\varepsilon^{1.5}}\right)$, with high constant probability, there is an island that approximates the optimal function in $\mathcal{C}_{conv}$ for distribution $\mathcal{D}$.

---

[1]As explained in Bautista-Santiago et al. (2011), "roughly speaking, a function $\alpha$ is decomposable if, when a polygon $P$ is cut into two subpolygons $P_1$ and $P_2$ along a diagonal $e$ joining vertices $p_1$ and $p_i$ of $P$, then $\alpha(P)$ can be calculated in constant time from $\alpha(P_1), \alpha(P_2)$, and some information on $e$."

---

**Algorithm 2:** Agnostic learner for convex sets in $[0,1]^2$ (with failure probability $1/3$)

---

**input** : loss parameter $\varepsilon \in (0,1)$; access to examples from a distribution $\mathcal{D}$ over $[0,1]^2 \times \{0,1\}$ with the uniform marginal distribution over $[0,1]^2$

1 Let $c_1$ and $c_2$ be large enough constants.

2 Sample a set $S$ of $s = \frac{c_1}{\varepsilon^{2.5}} \ln \frac{1}{\varepsilon}$ examples $(x_1, y_1), \ldots, (x_s, y_s)$ i.i.d. from $\mathcal{D}$.

3 Sample a set $N$ of $n = \frac{c_2}{\varepsilon^{1.5}}$ points u.a.r. from $[0,1]^2$.

4 Use algorithms $\mathcal{B}_{\text{pre}}$ and $\mathcal{B}_{\text{query}}$ from Theorem 2.3 to build a data structure on set $S$ and query it on each triangle $T \in \mathcal{T}_N$ to compute the asymmetric discrepancy $disc_S(T)$.

5 Run algorithm OptIsland from Theorem D.2 on the set $N$ with $\alpha(T) \stackrel{\text{def}}{=} disc_S(T)$ for all triangles $T \in \mathcal{T}_N$ to find an island $\tilde{I} \subseteq N$ with maximum asymmetric discrepancy $disc_S(\tilde{I})$.    \\Find the best island in $\mathcal{I}_N$.

6 Return the indicator function $f_{\tilde{I}}$.                                    \\See Definition B.4.

---

**Definition 3.4** (Optimal convex set $K$ and random set $K_n$). Let $K$ be the (geometric) convex set with the smallest error, i.e., $K = \arg\min_{C : f_C \in \mathcal{C}_{\text{conv}}} \{err_{\mathcal{D}}(f_C)\}$. For each $n \in \mathbb{N}$, let $K_n$ denote the convex hull of the set of points in a sample $N$ of size $n$ that fall inside $K$.

Since $K$ is a convex set, $K \cap N$ is an island in $\mathcal{I}_N$. Algorithm 2 considers all islands in $\mathcal{I}_N$, including $K \cap N$. We show that the polygon $K_n$ (defined by the island $K \cap N$) has error similar to that of $K$ w.r.t. to the uniform distribution over examples.

**Lemma 3.5.** If $n \geq \frac{c_2}{\varepsilon^{1.5}}$, then $|err_{\mathcal{D}}(f_K) - err_{\mathcal{D}}(f_{K_n})| \leq \frac{\varepsilon}{3}$ with probability at least $0.9$.

The absolute difference in the error of $f_K$ and $f_{K_n}$ is bounded above by the "missing area" between $K$ and $K_n$. The measure of the missing area is with respect to the uniform distribution. As the number of samples $n$ tends to infinity, the random variable $\mu(K \setminus K_n)$ goes to zero, as quantified in the following claim.

**Claim 3.6.** For a set $A \subseteq [0,1]^2$, let $\mu(A) = \Pr_{x \sim [0,1]^2}[x \in A]$. There exists a constant $C$ such $\Pr\left[\mu(K \setminus K_n) \geq Cn^{-2/3}\right] \leq 0.1$ for sufficiently large $n$.

*Proof.* Assume $n$ is sufficiently large. When $\mu(K) = O(n^{-2/3})$, the missing area never exceeds $Cn^{-2/3}$. Now assume $\mu(K) = \Omega(n^{-2/3})$. Let $L = |N \cap K|$, i.e., the random variable equal to the number of points that fall in $K$ among $n$ points selected u.a.r. from $[0,1]^2$. Observe that $N \cap K$ can be viewed as a uniform sample of $L$ points from $K$. Let random variable $\mu_L$ be $\mu(K \setminus K_n)$, that is, the measure of the "missing" area between $K$ and $K_n$. The value of this random variable depends on the number of points from the sample of size $n$ that fall inside $K$. For each $\ell \in \mathbb{N}$, let $\mu_\ell$ represent $\mu_L$ conditioned on $L = \ell$, i.e., on the event that $\ell$ points from a sample of size $n$ are in $K$. Then $\mu_\ell$ has the same distribution as the measure of the region between $K$ and the convex hull of $\ell$ points selected uniformly and independently at random from $K$.

The behavior of $\mu_\ell$, the missing area between a convex set and the convex hull of a uniform sample of size $\ell$ inside the set, has been extensively studied (see, e.g., Har-Peled (2011) and the survey in Bárány (2007)). By (Brunel, 2020, Theorem 4), there exist constants $\aleph_1, \aleph_2, \aleph_3$, such that for every convex body $K$ and $\ell$ points sampled uniformly and independently from $K$, for all $a \geq 0$, we have $\Pr\left[\frac{\mu_\ell}{\mu(K)} > \aleph_1 \cdot \ell^{-2/3} + \frac{a}{\ell}\right] \leq \aleph_2 \cdot e^{-\aleph_3 \cdot a}$. Therefore, for $a = \aleph_1 \cdot \ell^{1/3}$ and a sufficiently large $\ell$,

$$\Pr\left[\mu_\ell > 2\aleph_1 \cdot \mu(K) \cdot \ell^{-2/3}\right] \leq \aleph_2 \cdot e^{-\aleph_3 \cdot \aleph_1 \cdot \ell^{1/3}} \leq 0.05. \tag{1}$$

Now we analyze the behavior of the missing area $\mu_L$ when the number of samples that fall within the convex body $K$ is a random variable. Random variable $L$ has binomial distribution and expectation $\mathbb{E}[L] = n \cdot \mu(K)$. By the Chernoff bound, for sufficiently large $n$,

$$\Pr[L \leq 0.5n \cdot \mu(K)] \leq \Pr[L \leq 0.5 \cdot \mathbb{E}[L]] \leq e^{-n \cdot 0.5^2/2} = e^{-n/8} \leq 0.05. \tag{2}$$

Let $\ell_* = 0.5n \cdot \mu(K)$ and $Q = 2\aleph_1 \cdot \mu(K) \cdot \ell_*^{-2/3}$. By the law of total probability,

$$\begin{aligned}
\Pr[\mu_L \geq Q] &= \Pr[\mu_L \geq Q \mid L \geq \ell_*] \cdot \Pr[L \geq \ell_*] + \Pr[\mu_L \geq Q \mid L < \ell_*] \cdot \Pr[L < \ell_*] \\
&\leq \Pr[\mu_L \geq Q \mid L \geq \ell_*] \cdot 1 + 1 \cdot \Pr[L < \ell_*] \\
&\leq \Pr[\mu_L \geq Q \mid L = \ell_*] + \Pr[L < \ell_*] \tag{3} \\
&= \Pr[\mu_{\ell_*} \geq Q] + \Pr[L < \ell_*] \leq 0.1, \tag{4}
\end{aligned}$$

where (3) holds because $\Pr[\mu_L \geq Q \mid L = \ell]$ is monotonically decreasing in $\ell$, the equality in (4) holds by definition of $\mu_\ell$, and the inequality in (4) holds by (1) and (2). Note that we can use (1) because $n$ is sufficiently large and $\mu(K) = \Omega(n^{-2/3})$, implying that $\ell_*$ is sufficiently large.

We substitute $Q$ and $\ell_*$ into $\Pr[\mu_L \geq Q]$ and use (4) to obtain that with probability at least 0.9, $\mu_L < Q = 2\aleph_1 \cdot \mu(K) \cdot \ell_*^{-2/3} = 2\aleph_1 \cdot \mu(K) \cdot (0.5n \cdot \mu(K))^{-2/3} \leq Cn^{-2/3}$, where $C = 2^{5/2}\aleph_1$ and the last inequality holds because $\mu(K) \leq 1$. Recall that $\mu_L = \mu(K \setminus K_n)$, completing the proof of Claim 3.6. $\qquad\square$

*Proof of Lemma 3.5.* By Claim 3.6, with probability at least 0.9, $|err_{\mathcal{D}}(f_K) - err_{\mathcal{D}}(f_{K_n})| \leq \Pr_{(x,y) \in \mathcal{D}}[x \in K \setminus K_n] \leq \frac{C}{n^{2/3}} \leq \frac{\varepsilon}{3}$, where the last inequality holds since $n = \frac{c_2}{\varepsilon^{1.5}}$ for sufficiently large constant $c_2$. $\qquad\square$

### 3.2 EMPIRICAL RISK IS ACCURATE FOR ALL ISLANDS

In this section, we show that sample $S$ is likely to be representative for all islands in $\mathcal{I}_N$. We rely on Valtr's theorem (formally stated in Theorem D.4 in Appendix D.3) in computational geometry that implies that, for some constant $\lambda$, with high probability, all islands have at most $\lambda n^{1/3}$ vertices in their convex hull. See Figure 2 for an illustration of vertices of a convex hull of an island. We then use Theorem D.4 to show that with large enough probability, we get a sample $S$ which is $\varepsilon$-representative for for all islands induced by $N$. (Definition B.8 recalls what $\varepsilon$-representative means).

**Lemma 3.7** (Sample $S$ is an $\frac{\varepsilon}{3}$-representative for islands). *Let $c_1$ be a sufficiently large constant and $N$ be a sample of $n$ points drawn uniformly and independently at random from $[0,1]^2$. A sample $S$ of size $s = \frac{c_1}{\varepsilon^2} \cdot n^{1/3} \ln n$ drawn i.i.d. from $\mathcal{D}$ is an $\frac{\varepsilon}{3}$-representative for all islands in $\mathcal{I}_N$ with probability at least $\frac{9}{10}$. (The probability is taken over both samples, $N$ and $S$.)*

*Proof.* By Theorem D.4, for sufficiently large $n$, in a uniform sample $N$ of size $n$, with probability at least $\frac{19}{20}$, the maximum number of vertices in the convex hull of any island $I$ in $\mathcal{I}_N$ is at most $7n^{1/3}$. Condition on this event. Let $\mathcal{H}$ be the concept class of all polygons corresponding to convex hulls of islands in $\mathcal{I}_N$. By the conditioning, all these polygons have at most $7n^{1/3}$ vertices. Thus, the size of this concept class is $|\mathcal{H}| = \sum_{k=1}^{7n^{1/3}} \binom{n}{k} \leq 7n^{1/3} \cdot \binom{n}{7n^{1/3}} \leq 7n^{1/3} (\frac{ne}{7n^{1/3}})^{7n^{1/3}}$. By the uniform convergence bound for finite classes (Appendix B.4) a set $S$ of size $m_{\mathcal{C}}(\frac{\varepsilon}{3}, \frac{1}{20}) = \frac{c_1 \log |\mathcal{H}|}{\varepsilon^2} = \frac{c_1 \cdot n^{1/3} \ln n}{\varepsilon^2}$ (for a sufficiently large constant $c_1$) is $\frac{\varepsilon}{3}$-representative for $\mathcal{H}$ with probability at least $\frac{19}{20}$.

The lemma follows from taking a union bound over the two failure events: having a convex hull of an island with more than $7n^{1/3}$ vertices and failing to have a representative set. $\qquad\square$

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

## A   IMPLICATIONS FOR PROPERTY TESTING

Agnostic learning is tightly related to computational tasks first investigated in by Parnas et al. (2006) and studied in property testing: distance approximation and tolerant testing.

**Definition A.1** (Distance Approximation). Let $\mathcal{C}$ be a class of functions $f : \mathcal{X} \to \mathcal{Y}$ (also referred to as a *property*). A *distribution-free distance approximation* algorithm $\mathcal{A}$ for $\mathcal{C}$ is given sample access to an unknown distribution $\mathcal{D}_{\mathcal{X}}$ on the domain $X$ and query access to an input function $f : \mathcal{X} \to \mathcal{Y}$, as well as parameters $\varepsilon, \delta \in (0, 1)$. Let $\mathcal{D}$ denote the joint distribution on $(x, y) \in \mathcal{X} \times \mathcal{Y}$, where $x \sim \mathcal{D}_{\mathcal{X}}$ and $y = f(x)$. Algorithm $\mathcal{A}$ must return a number $\hat{d}$ such that $dist_{\mathcal{D}}(f) - \varepsilon \leq \hat{d} \leq dist_{\mathcal{D}}(f) + \varepsilon$ with probability at least $1 - \delta$. The query complexity of $\mathcal{A}$ is the number of queries it makes to $f$ in the worst case over the choice of $\mathcal{D}_{\mathcal{X}}$ and $f \in \mathcal{C}$.

A *tolerant tester* gets the same inputs as the distance approximation algorithm, with an additional parameter $\varepsilon_0 \in (0, \varepsilon)$, and it has to accept if $dist_{\mathcal{D}}(f) \leq \varepsilon_0$ and reject if $dist_{\mathcal{D}}(f) \geq \varepsilon$, both with probability at least $1 - \delta$. Distance approximation and tolerant testing have closely related query complexity, as stated, for example, by Parnas et al. (2006) and (Pallavoor et al., 2022, Theorem 5.1). Most work in property testing considers the special case of the distribution-free version of these problems, where the marginal distribution $\mathcal{D}_{\mathcal{X}}$ is fixed to be uniform over $\mathcal{X}$.

As proved by Goldreich et al. (1998), a proper PAC learning algorithm for a class $\mathcal{C}$ with sample complexity $s(\varepsilon)$ implies a tester for property $\mathcal{C}$ that makes $s(\varepsilon/2) + O(1/\varepsilon)$ queries to the input. There is an analogous implication from proper agnostic PAC learning to distance approximation with an additive overhead of $O(1/\varepsilon^2)$ instead of $O(1/\varepsilon)$. Therefore, our agnostic PAC learners imply distance approximation algorithms with the same sample and time complexity. Specifically, we can estimate the distance to the nearest $k$-gon with constant error probability in time $\tilde{O}(\frac{1}{\varepsilon^{2k}})$ and the distance to the nearest convex set in time $\tilde{O}(\frac{1}{\varepsilon^5})$.

## B   PRELIMINARIES

### B.1   AGNOSTIC LEARNING

The *agnostic learning* framework of Haussler (1992); Kearns et al. (1994) models learning from noisy data. In this framework, a learning algorithm $\mathcal{A}$ is given examples of the form $(x, y) \in \mathcal{X} \times \mathcal{Y}$, where $\mathcal{X}$ represents

a domain (e.g., $\mathbb{R}^d$ or $[n]^d$ for some $d \in \mathbb{N}$), and $\mathcal{Y}$ is the set of labels (typically, $\{0,1\}$). The examples are drawn i.i.d. from some unknown distribution $\mathcal{D}$ on $\mathcal{X} \times \mathcal{Y}$. A *concept* is a function $f : \mathcal{X} \to \mathcal{Y}$. In contrast to the PAC learning framework of Valiant (1984), where there is some underlying concept $f$ producing the labels, i.e., $y = f(x)$, in agnostic learning, the labels come from the distribution. A *concept class* $\mathcal{C}$ is a set of concepts. The goal of $\mathcal{A}$ is to output a *hypothesis* $h$ in a specified concept class $\mathcal{C}$. A concept $h$ is *consistent* with an example $(x, y)$ if $h(x) = y$; otherwise, $h$ *mislabels* the example. The *error* of a concept $h$ is measured with respect to the distribution $\mathcal{D}$: specifically, $err_{\mathcal{D}}(h) = \Pr_{(x,y)\sim\mathcal{D}}[h(x) \neq y]$, i.e., the probability that a random example drawn from $\mathcal{D}$ is mislabeled by $h$. The smallest possible error, denoted $OPT$, is $\min_{f \in C}\{err_{\mathcal{D}}(f)\}$. The algorithm is given two parameters: the *loss parameter* $\varepsilon \in (0, 1)$, specifying how much the error of the output hypothesis is allowed to deviate from $OPT$, and the failure probability parameter $\delta \in (0, 1)$. The number of examples $\mathcal{A}$ draws from $\mathcal{D}$ (in the worst case over $\mathcal{D}$) is denoted $m(\varepsilon, \delta)$ and is called the *sample complexity* of $\mathcal{A}$.

**Definition B.1** (Agnostic PAC learning). Let $\mathcal{C}$ be a class of concepts $f : \mathcal{X} \to \mathcal{Y}$. An algorithm $\mathcal{A}$ is an *agnostic PAC learner* for $\mathcal{C}$ with sample complexity $m(\varepsilon, \delta)$ if, for every joint distribution $\mathcal{D}$ over $\mathcal{X} \times \mathcal{Y}$, given *loss parameter* $\varepsilon \in (0, 1)$ and *failure probability parameter* $\delta \in (0, 1)$, algorithm $\mathcal{A}$ draws $m(\varepsilon, \delta)$ examples i.i.d. from $\mathcal{D}$ and returns a hypothesis $h$ such that

$$\Pr[err_{\mathcal{D}}(h) \leq \min_{f \in C}\{err_{\mathcal{D}}(f)\} + \varepsilon] \geq 1 - \delta,$$

where the probability is taken over $\mathcal{D}$ and the coins of $\mathcal{A}$. The learner $\mathcal{A}$ is *proper* if it always returns a hypothesis in $\mathcal{C}$.

We measure the running time of the algorithm using the RAM model, where each basic arithmetic operation and memory access can be performed in a single step.

## B.2 GEOMETRIC PROPERTIES

We start by defining geometric properties we consider.

**Definition B.2** (The class of halfplanes). A *halfplane* is an indicator function $f_{a,b} : \mathbb{R}^2 \to \{0, 1\}$ indexed by $a \in \mathbb{R}^2, b \in \mathbb{R}$, where $f_{a,b}(x) = 1$ iff $a^T x \leq b$. The set of all halfplanes in $\mathbb{R}^2$ is denoted $\mathcal{H}^2$.

Given a set of points $P$, let $\mathrm{Hull}(P)$ represent the convex hull of $P$.

**Definition B.3** (The class of $k$-gons). Let $\mathcal{X} \subseteq \mathbb{R}^2$. A *$k$-gon* over $\mathcal{X}$ is an indicator function $f_P : \mathcal{X} \to \{0, 1\}$ indexed by a set $P \in \mathbb{R}^2$ of $k$ points in general position, where $f_P(x) = 1$ iff $x \in \mathrm{Hull}(P)$. The set of all $k$-gons over $\mathcal{X}$ is denoted $\mathcal{C}_k$.

**Definition B.4** (The class of convex sets). Let $\mathcal{X} \subseteq \mathbb{R}^2$. A *convex set* over $\mathcal{X}$ is an indicator function $f_P : \mathcal{X} \to \{0, 1\}$ indexed by a (finite or infinite) set of points $P \subseteq \mathbb{R}^2$, where $f_P(x) = 1$ iff $x \in \mathrm{Hull}(P)$. The set of all convex sets over $\mathcal{X}$ is denoted $\mathcal{C}_{\mathrm{conv}}$.

For a function $f : \mathbb{R}^d \to \{0, 1\}$, let $f^{-1}(1)$ denote the set of points $x \in \mathbb{R}^d$ on which $f$ evaluates to 1, i.e., $f^{-1}(1) = \{x \in \mathbb{R}^d : f(x) = 1\}$. If $f$ is an indicator function for some set $P$ then $f^{-1}(1) = P$.

## B.3 EMPIRICAL RISK AND DISCREPANCY

We use standard definitions of empirical risk (also called empirical error) and other notions from learning theory (see, e.g., the textbook by Shalev-Shwartz & Ben-David (2014)).

**Definition B.5.** The *empirical risk* of a concept $h : \mathcal{X} \to \mathcal{Y}$ on a set $S \subseteq \mathcal{X} \times \mathcal{Y}$ of examples is $err_S(h) = \frac{1}{|S|}\big|\{(x, y) \in S : h(x) \neq y\}\big|$, i.e., the fraction of examples in $S$ mislabeled by $h$.

Next, we define asymmetric discrepancy of a polygon w.r.t. a sample of points.

**Definition B.6** (Asymmetric discrepancy). Fix a polygon $P$ and a multiset $S \subset \mathbb{R}^2 \times \{0, 1\}$ of labeled examples $\{x_i, y_i\}_{i \in S}$. The *asymmetric discrepancy* of $P$ (w.r.t. $S$) is

$$disc_S(P) = |\{x \in P \cap S : y = 1\}| - |\{x \in P \cap S : y = 0\}|.$$

It is well known (see, e.g., Lemma 2 in Fischer (1995)) that the asymmetric discrepancy of a polygon $P$ is related to the empirical risk of its indicator function $f_P$. For completeness, we state the relationship in the following claim.

**Claim B.7** (Asymmetric discrepancy vs. empirical risk). *Fix a polygon $P$ and a multiset $S \subset \mathbb{R}^2 \times \{0, 1\}$ of labeled examples. Let $S^+$ be the set of positive examples, i.e., $S^+ = \{(x, y) \in S : y = 1\}$. Then $disc_S(P) = |S^+| - err_S(f_P) \cdot |S|$.*

*Proof.* By definition of the asymmetric discrepancy,

$$\begin{aligned}
disc_S(P) &= |\{x \in P \cap S : y = 1\}| - |\{x \in P \cap S : y = 0\}| \\
&= |\{x \in S : y = 1\}| - |\{x \in S \setminus P : y = 1\}| - |\{x \in P \cap S : y = 0\}| \\
&= |\{x \in S : y = 1\}| - |\{x \in S : f_P(x) \neq y\}| \\
&= |S^+| - err_S(f_P) \cdot |S|,
\end{aligned}$$

where the last equality is obtained from the definition of $S^+$ and the empirical risk. $\square$

Since $S^+$ and $S$ do not depend on the polygon, Claim B.7 implies that maximizing the discrepancy over some set of polygons is equivalent to minimizing the empirical risk of their indicator functions.

### B.4 UNIFORM CONVERGENCE

**Definition B.8** ($\varepsilon$-representative set of examples). A set $S \subseteq \mathcal{X} \times \mathcal{Y}$ of examples is called $\varepsilon$-representative for hypothesis class $\mathcal{C}$ w.r.t. distribution $\mathcal{D}$ if for all $f \in \mathcal{C}$,

$$|err_S(f) - err_{\mathcal{D}}(f)| \leq \varepsilon.$$

**Definition B.9** (Uniform convergence). A hypothesis class $\mathcal{C}$ has the uniform convergence property if there exists a function $m_{\mathcal{C}}^{UC} : \{0, 1\}^2 \to \mathbb{N}$ such that for every $\varepsilon, \delta \in (0, 1)$ and for every probability distribution $\mathcal{D}$ over $\mathcal{X} \times \mathcal{Y}$, if $S$ is a sample of $m \geq m_{\mathcal{C}}^{UC}(\varepsilon, \delta)$ examples drawn i.i.d. from $\mathcal{D}$, then, with probability of at least $1 - \delta$, sample $S$ is $\varepsilon$-representative for $\mathcal{C}$ w.r.t. $\mathcal{D}$. In this case, we say that $m_{\mathcal{C}}^{UC}(\varepsilon, \delta)$ examples are sufficient to get uniform convergence for $\mathcal{C}$ with loss parameter $\varepsilon$ and failure probability $\delta$. If the requirement is satisfied for one specific distribution $\mathcal{D}$, as opposed to all $\mathcal{D}$, we refer to it as the uniform convergence w.r.t. $\mathcal{D}$.

By the Fundamental Theorem of Statistical Learning (Shalev-Shwartz & Ben-David, 2014, Theorems 6.7-6.8), $m_{\mathcal{C}}(\varepsilon, \delta) = O(\frac{1}{\varepsilon^2}(\text{VC-dim}(\mathcal{C}) + \ln \frac{1}{\delta}))$ examples sampled i.i.d. from distribution $\mathcal{D}$ are sufficient to get uniform convergence for $\mathcal{C}$ and to agnostically PAC learn $\mathcal{C}$. By (Shalev-Shwartz & Ben-David, 2014, Corollary 4.6), for the special case when the concept class $\mathcal{C}$ is finite, $m_{\mathcal{C}}(\varepsilon, \delta) = O(\frac{1}{\varepsilon^2}(\ln |\mathcal{C}| + \ln \frac{1}{\delta}))$. A hypothesis from $\mathcal{C}$ that minimizes empirical risk is abbreviated as ERM. Any algorithm that gets $m \geq m_{\mathcal{C}}(\frac{\varepsilon}{2}, \frac{\delta}{2})$ examples and outputs an ERM hypothesis is an agnostic PAC learner for $\mathcal{C}$. Finally, an algorithm that gets that many samples and outputs a hypothesis that has empirical risk within $\frac{\varepsilon}{4}$ of an ERM is also an agnostic PAC learner.

## C MATERIAL DEFERRED FROM SECTION 2

The following figures illustrate concepts from Section 2.

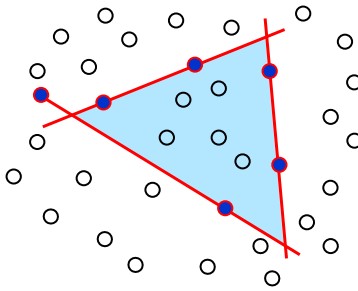

Figure C.1: An illustration to Definition 2.5: a set of points and a (light blue) reference triangle it defines.

Figure C.2: An illustration to Lemma 2.6: a (waved) triangle $t$ and a (light blue) nearby reference triangle.

## D  MATERIAL DEFERRED FROM SECTION 3

### D.1  MISSING FIGURES

The following figures were deferred from Section 3.

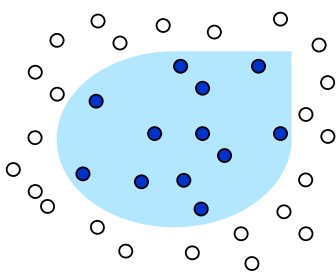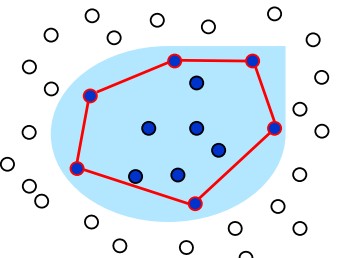

Figure 1: An illustration of an island: a set of points is intersected with a (light blue) convex set; the (blue) points in the intersection form an island.

Figure 2: The convex hull of the island is delineated by (red) lines; the (six) vertices of the convex hull have the red border.

### D.2  DECOMPOSABLE FUNCTIONS AND THE OPTISLAND ALGORITHM

Recall that our algorithm uses a data structure of Bautista-Santiago et al. (2011) which relies on the notion of decomposable functions, defined next.

**Definition D.1** (Definition 1 in Bautista-Santiago et al. (2011)). Let $\mathcal{P}$ be the set of all convex polygons. A function $\alpha : \mathcal{P} \to \mathbb{R}$ is decomposable if there is a constant-time computable function $\beta\colon \mathbb{R} \times \mathbb{R} \times \mathbb{R}^2 \times \mathbb{R}^2 \to \mathbb{R}$ such that, for any polygon $P = \mathrm{Hull}(p_1, p_2, ..., p_k) \in \mathcal{P}$ and any index $2 < i < k$, it holds that $\alpha(P) = \beta(\alpha(\mathrm{Hull}(p_1, ..., p_i)), \alpha(\mathrm{Hull}(p_1, p_i, ..., p_k)), p_1, p_i)$. The function $\alpha$ is *monotone* if $\beta$ is monotone in the first and second argument.

**Theorem D.2** (Theorem 3 in Bautista-Santiago et al. (2011)). *Let $N$ be a set of $n$ points in general position in the plane and let $\alpha : \mathcal{P} \to \mathbb{R}$ be a monotone decomposable function. There exists an algorithm, OptIslands, that computes the island that minimizes (maximizes) $\alpha$ in $O(n^3 + B(n))$ time, where $B(n)$ is the time required to compute $\alpha$ for the $O(n^3)$ triangles induced by the set $N$.*

To invoke the OptIslands algorithm from Theorem D.2, we define $\alpha(P) = disc_S(P)$ for every convex polygon $P$, where $S$ is a sample drawn from $\mathcal{D}$.

**Observation D.3.** *Let $P$ be a polygon with vertices $p_1, \ldots, p_k$. If points in $N \cup S$ are in general position, then $\alpha(P) = \alpha(Hull(p_1, \ldots, p_i)) + \alpha(Hull(p_1, p_i, \ldots, p_k))$, so $\alpha$ is decomposable and monotone.*

### D.3 VALTR'S THEOREM

**Theorem D.4** (Valtr (1994; 1995)). *Let $N$ be a set of $n$ points chosen independently and uniformly from the unit square. For every polygon $P$, let $\nu(P)$ denote the number of vertices in $P$. Let $V = \max_{I \in \mathcal{I}_N} \nu(P_I)$, that is, the largest number of vertices in the convex hull of an island. Let $h = 2^{4/3}e \approx 6.85$. Then $\Pr[V > \lambda \cdot n^{1/3}] < \left(\frac{h}{\lambda}\right)^{3\lambda n^{1/3}}$ for all $\lambda \geq h$.*

### D.4 THE PROOF OF THEOREM 3.2

*Proof of Theorem 3.2.* First, observe that $s$ points drawn from the uniform distribution over $[0,1]^2$ are in general position with probability 1. Thus, the invocation of OptIsland in Algorithm 2 works correctly with probability 1. Next we analyze the failure probability and the loss of Algorithm 2. Algorithm 2 returns hypothesis $f_{\tilde{I}}$, where $\tilde{I}$ is the island with maximum asymmetric discrepancy and, hence, by Claim B.7, with smallest empirical risk. Recall that $K$ denotes the convex set such that $err_{\mathcal{D}}(f_K) = OPT$, where $OPT = \min_{f \in \mathcal{C}_{\text{conv}}} err_{\mathcal{D}}(f)$. Also recall that $K_n$ is the convex hull of the points from sample $N$ of size $n$ that fall inside $K$.

Consider the following three failure events: that the number of vertices in the largest island in $\mathcal{I}_N$ exceeds $7n^{1/3}$, that sample $S$ is not $\frac{\varepsilon}{3}$-representative, and that $|err_{\mathcal{D}}(f_K) - err_{\mathcal{D}}(f_{K_n})| > \frac{\varepsilon}{3}$. By Theorem D.4, the first event occurs with probability at most 0.1. By Lemmas 3.7 and 3.5, each of the latter two events occur with probability at most 0.1 for our setting of $s$ and $n$. (This is because $s = \frac{c_1}{\varepsilon^2} \cdot n^{1/3} \ln n$ and $n = \frac{c_2}{\varepsilon^{1.5}}$ for sufficiently large constants $c_1$ and $c_2$, and consequently $s = \frac{c_1}{\varepsilon^{2.5}} \ln \frac{1}{\varepsilon}$). Thus, by a union bound, one or more of these events happens with probability at most 0.3. If none of the failure events happened then

$$
\begin{aligned}
err_{\mathcal{D}}(f_{\tilde{I}}) &\leq err_S(f_{\tilde{I}}) + \frac{\varepsilon}{3} && s \text{ is } \frac{\varepsilon}{3}\text{-representative for } s = \frac{c_1}{\varepsilon^2} \cdot n^{1/3} \cdot \ln n \text{ (Lemma 3.7)} \\
&\leq err_S(f_{K_n}) + \frac{\varepsilon}{3} && \tilde{I} \text{ is the island that minimizes the empirical error} \\
&\leq err_{\mathcal{D}}(f_{K_n}) + \frac{2\varepsilon}{3} && \text{by Lemma 3.7} \\
&\leq err_{\mathcal{D}}(f_K) + \varepsilon && \text{by Lemma 3.5 for } n = \frac{c_2}{\varepsilon^{1.5}} \\
&= OPT + \varepsilon.
\end{aligned}
$$

Thus, we get that with probability at least $\frac{2}{3}$, the hypothesis $f_{\tilde{I}}$ returned by Algorithm 2 satisfies $err_{\mathcal{D}}(f_{\tilde{I}}) \leq OPT + \varepsilon$, and that $f_{\tilde{I}}$ is an indicator function for a convex polygon with at most $7n^{1/3} = O(\frac{1}{\varepsilon^{0.5}})$ vertices.

It remains to analyze the complexity. The sample complexity is $O(n + s) = O(s) = O(\frac{c_1}{\varepsilon^{2.5}} \ln \frac{1}{\varepsilon})$. By Theorems 2.3 and D.2, the running time is $O(s^2 + n^3 \log s) = O(\frac{1}{\varepsilon^5} \ln^2 \frac{1}{\varepsilon})$. $\qquad\square$

## E AMPLIFICATION OF SUCCESS PROBABILITY

In this section, we use standard arguments to amplify the success probability of the two learners to $1 - \delta$, for any given $\delta \in (0, \frac{1}{2})$. We prove Lemma E.1 as well as Theorems 2.1 and 3.1. Given a learner $\mathcal{A}$ with

constant success probability, Algorithm 3 calls it repeatedly and evaluates the hypotheses obtained from the calls on a fresh sample.

---

**Algorithm 3:** Amplified-Success-Learner

---

**input** : A loss parameter $\varepsilon \in (0, 1)$, a failure probability parameter $\delta \in (0, \frac{1}{2})$, a learner $\mathcal{A}$.

**1** Invoke the learner $\mathcal{A}$ independently $t = \ln \frac{2}{\delta}$ times with loss parameter $\frac{\varepsilon}{3}$. For $j \in [t]$, let $h_j$ denote the hypothesis returned by the $j^{\text{th}}$ invocation.

**2** Sample a set $Q$ of $q = \frac{9}{\varepsilon^2} \ln \frac{1}{\delta}$ examples i.i.d. from $\mathcal{D}$.

**3** For each $j \in [t]$, compute $err_Q(h_j)$.

**4** Return hypothesis $\hat{h} = \arg\min_{j \in [t]} \{err_Q(h_j)\}$.

---

**Lemma E.1.** *Let $\mathcal{C}$ be a class of concepts $f : \mathcal{X} \to \mathcal{Y}$ and $\mathcal{D}$ be a distribution on labeled examples. Let $\mathcal{A}$ be an agnostic PAC learner for $\mathcal{C}$ w.r.t. distribution $\mathcal{D}$ that has failure probability $\frac{1}{3}$ and sample complexity $S_{\mathcal{A}}$. Then Algorithm 3 is an agnostic PAC learner for $\mathcal{C}$ w.r.t. distribution $\mathcal{D}$ that takes failure probability $\delta \in (0, \frac{1}{2})$ as input and has sample complexity $O(S_{\mathcal{A}} \cdot \ln \frac{1}{\delta} + \frac{1}{\varepsilon^2} \ln \frac{1}{\delta})$.*

*Proof.* Fix some $h$ in $\{h_j\}_{j \in [t]}$. For $i \in [q]$, let $\chi_i$ be the indicator for the event that $h(x_i) \neq y_i$. Let $\chi = \sum_{i \in [q]} \chi_i$. Note that $\chi = q \cdot err_Q(h)$ and $\mathbb{E}[\chi] = q \cdot err_{\mathcal{D}}(h)$. By the Hoeffding bound,

$$\Pr\left[|err_Q(h) - err_{\mathcal{D}}(h)| \geq \frac{\varepsilon}{3}\right] \leq 2\exp\left(-2\left(\frac{\varepsilon}{3}\right)^2 \cdot q\right) = \frac{2}{\delta^2}.$$

We call sample $Q$ *representative* if $|err_Q(h) - err_{\mathcal{D}}(h)| \leq \frac{\varepsilon}{3}$ for all $h \in \{h_j\}_{j \in [t]}$. By a union bound over the $t = \ln \frac{2}{\delta}$ invocations and for $\delta \in (0, \frac{1}{2})$, the probability of $Q$ being representative is at least $1 - \frac{\delta}{2}$.

Let $h^*$ be $\arg\min_{j \in [t]} \{err_{\mathcal{D}}(h_j)\}$, i.e., the best hypothesis w.r.t. to $\mathcal{D}$ among $h_1, \dots, h_t$. Since algorithm $\mathcal{A}$ has failure probability $\frac{2}{3}$, we get $\Pr[err_{\mathcal{D}}(h_j) > OPT + \frac{\varepsilon}{3}] \leq \frac{1}{3}$ for all $j \in [t]$. We call $h^*$ *good* if $err_{\mathcal{D}}(h^*) \leq OPT + \frac{\varepsilon}{3}$. Since $h^*$ is the function that minimizes $err_{\mathcal{D}}(h_j)$ among the $t$ functions, $h^*$ is not good with probability at most $(\frac{1}{3})^t < \frac{\delta}{2}$. Therefore, with probability at least $1 - \delta$, sample $Q$ is representative and $h^*$ is good. We get

$$
\begin{aligned}
err_{\mathcal{D}}(\hat{h}) &\leq err_Q(\hat{h}) + \frac{\varepsilon}{3} & & Q \text{ is a representative sample} \\
&\leq err_Q(h^*) + \frac{\varepsilon}{3} & & \hat{h} \text{ minimizes empirical risk w.r.t. to } Q \\
&\leq err_{\mathcal{D}}(h^*) + \frac{\varepsilon}{3} + \frac{\varepsilon}{3} & & Q \text{ is a representative sample} \\
&\leq OPT + \varepsilon\,, & & h^* \text{ is good}
\end{aligned}
$$

as stated.

The sample complexity is due to the sampling of the set $Q$ and the $O(\ln \frac{1}{\delta})$ invocations of the learner, yielding the overall sample complexity of $O(\frac{1}{\varepsilon^2} \ln \frac{1}{\delta} + \ln \frac{1}{\delta} S_{\mathcal{A}})$. $\qquad \square$

We are now ready to prove Theorem 2.1.

*Proof of Theorem 2.1.* Let $\mathcal{A}$ be an algorithm that first takes a sample $S$ of size $\frac{c}{\varepsilon^2}$ for some sufficiently large constant $c$, so that with probability at least $\frac{5}{6}$, the sample $S$ is $\varepsilon$-representative for $\mathcal{C}_k$, and then invokes

Algorithm 1 with the set $S$ and loss parameter $\varepsilon$. Then by the discussion prior to Theorem 2.2, Algorithm $\mathcal{A}$ is an $(\varepsilon, \frac{2}{3})$-agnostic PAC learner for $\mathcal{C}_k$ with sample complexity $S_\mathcal{A} = O(\frac{1}{\varepsilon^2})$ and running time $T_\mathcal{A} = O(\frac{1}{\varepsilon^{2k}} \ln \frac{1}{\varepsilon} + \frac{1}{\varepsilon^4}) = O(\frac{1}{\varepsilon^{2k}} \ln \frac{1}{\varepsilon})$.

Therefore, by Lemma E.1, invoking Algorithm 3 with $\mathcal{A}$ results in an $(\varepsilon, \delta)$-agnostic PAC learner for $\mathcal{C}_k$ with sample sample complexity $O(S_\mathcal{A} \cdot \ln \frac{1}{\delta} + \frac{1}{\varepsilon^2} \ln \frac{1}{\delta}) = O(\frac{1}{\varepsilon^2} \ln \frac{1}{\delta})$. For the running time analysis observe that for every constant $k$, hypothesis $h \in \mathcal{C}_k$, and $x \in \mathbb{R}^2$, computing $h(x)$ takes $O(1)$ time. Therefore, the running time is $O(T_\mathcal{A} \cdot \ln \frac{1}{\delta} + \frac{1}{\varepsilon^2} \ln^2 \frac{1}{\delta}) = O(\frac{1}{\varepsilon^{2k}} \ln \frac{1}{\varepsilon} \ln \frac{1}{\delta} + \frac{1}{\varepsilon^2} \ln^2 \frac{1}{\delta})$. $\qquad \square$

In the proof of Theorem 3.1, we use the following algorithm in order to efficiently evaluate the error of the $t$ hypotheses on the sample $Q$.

**Theorem E.2** (Theorem 1 in Brodal & Jacob (2002)). *There exists an algorithm that, given a convex polygon $P$ in the plane with $v$ vertices and a set $Q$ of $q$ points on the plane, returns the set of points $P \bigcap Q$ in time $O((v + q) \log v)$.*

*Proof of Theorem 3.1.* Let $\mathcal{A}$ be the algorithm that runs Algorithm 2 and outputs the hypothesis $f_{\tilde{I}}$ returned by Algorithm 2 if $\text{Hull}(\tilde{I})$ has $O(\varepsilon^{-0.5})$ vertices and fails otherwise. Note that $\mathcal{A}$ has the same guarantees as those of Algorithm 2, with the additional guarantee that it always outputs an indicator function for a bounded size polygon or fails. Hence, it has sample complexity $S_\mathcal{A} = O(\frac{1}{\varepsilon^{2.5}} \ln \frac{1}{\varepsilon})$ and running time $T_\mathcal{A} = O(\frac{1}{\varepsilon^5} \ln^2 \frac{1}{\varepsilon})$.

Now we analyze the complexity of Algorithm 3 invoked with $\mathcal{A}$.

In order to efficiently compute the error empirical error of the $t$ hypothesis on the set $Q$ we do as follows. For each hypothesis that is an indicator function for some polygon $P$ over $\nu(P) = c\varepsilon^{-0.5}$ vertices and for the $Q$ examples, we invoke the algorithm described in Theorem E.2 to compute $P \bigcap Q$. This takes time $O((q + \nu(P)) \cdot \log |\nu(P)|) = O(\frac{1}{\varepsilon^2} \ln \frac{1}{\varepsilon} \ln \frac{1}{\delta})$ for each hypothesis. Therefore, computing the empirical error of all $t = O(\ln \frac{1}{\delta})$ hypotheses takes time $O(\frac{1}{\varepsilon^2} \ln \frac{1}{\varepsilon} \ln^2 \frac{1}{\delta})$.

Thus, the sample complexity of the $(\varepsilon, \delta)$-learner for $\mathcal{C}_{\text{conv}}$ is $O(S_\mathcal{A} \cdot \frac{1}{\delta} + \frac{1}{\varepsilon^2} \ln \frac{1}{\delta}) = O(\frac{1}{\varepsilon^{2.5}} \ln \frac{1}{\varepsilon} \ln \frac{1}{\delta} + \frac{1}{\varepsilon^2} \ln \frac{1}{\delta}) = O(\frac{1}{\varepsilon^{2.5}} \ln \frac{1}{\varepsilon} \cdot \ln \frac{1}{\delta})$, and the running time is $O(t \cdot T_\mathcal{A} + \frac{1}{\varepsilon^2} \ln \frac{1}{\varepsilon} \ln \frac{1}{\delta}) = O(\frac{1}{\varepsilon^5} \ln^2 \frac{1}{\varepsilon} \cdot \ln \frac{1}{\delta} + \frac{1}{\varepsilon^2} \ln \frac{1}{\varepsilon} \ln^2 \frac{1}{\delta})$.

Finally, since by Theorem 3.2, Algorithm 2 is an $(\varepsilon, \frac{2}{3})$-agnostic PAC learner for the class $\mathcal{C}_{\text{conv}}$ under every distribution $\mathcal{D}$ over $\mathcal{X} \times \{0, 1\}$, such that $\mathcal{D}_\mathcal{X}$ is uniform, so is algorithm $\mathcal{A}$, and therefore, Algorithm 3 invoked with $\mathcal{A}$ is an $(\varepsilon, \delta)$ learner for $\mathcal{C}_{\text{conv}}$ under the same distribution. $\qquad \square$

