# OpenReview forum: "Fast Agnostic Learners in the Plane"
_ICLR.cc/2026/Conference — ICLR 2026 Conference Withdrawn Submission_

### Official Review · Reviewer_5k1A · 2025-10-24

**Soundness:** 3
**Presentation:** 3
**Contribution:** 3
**Rating:** 6
**Confidence:** 2

**Summary:**

This paper introduces two new algorithms for proper agnostic learning in the plane. The first is for learning k-gons over any distribution, and the second is for learning arbitrary convex sets in the plane over the uniform distribution. Both algorithms go through a two stage process of first coming up with a (small) set of candidate hypotheses, and then running ERM over this restricted set.

One cool insight of prior work is that computing the empirical risk of a k-gon hypothesis is equivalent to the so-called "asymmetric discrepancy” problem from computational geometry, which asks for the difference between the number of positive and negative labels inside a k-gon. Using this insight, we can triangulate any polygon, and then compute the discrepancy of each triangle, aggregating over all triangles in order to compute the empirical risk of each k-gon. We do this for each k-gon in a small “reference set” of k-gons, and then return the k-gon with the best error. The reference set of k-gons is constructed by sampling a small number N of points from the distribution, and then considering all intersections of k halfspaces generated by selecting pairs of points from N. The analysis, based on extending the prior work of Matheny and Phillips (2018b), shows that this turns out to be a reference set (essentially, an $\epsilon$-net on the hypothesis class of $k$-gons). The remaining analysis seems to be standard tools for learning theory.

The proposed algorithm for general convex sets is also quite clever. It samples a small set N from the distribution, and then considers “islands”, which are convex sets intersected with N. By using randomized islands (via the set N), a lot of painful analysis and computational geometry can be avoided. Combined with a concentration result of Brunel (2020), this demonstrates that there exists a (randomized) island which well-approximates the optimal convex set, which we return.

A key element of both proposed algorithms seems to be the fast data structure of Goswami et al. (2004) used to calculate the asymmetric disparity of triangles.

**Strengths:**

Note that I do not have any experience in computational geometry (CG), and it seems that most of the proof techniques involved rely on recent and prior data structures or algorithms in CG. I do not have any intuition as to whether these algorithms will work for the tasks at hand, or if they are reasonable tools for the job. Therefore, I cannot feasibly check most of the proofs given the constrained reviewing period.

That being said, I did enjoy reading and learning about the area through the paper. In terms of the major strengths:
1. The sample and runtime complexity for the proposed algorithms are clearly significantly better than prior work, which seems to be almost 30 years old.
2. The paper is relatively readable for someone with no experience in computational geometry.

**Weaknesses:**

My only major concern is whether the audience of ICLR would find this work interesting, or if it should be submitted to a more learning theory oriented conference such as COLT / ALT or a computational geometry conference. I don’t have a strong opinion either way, but I can see the authors of the submitted paper getting less engagement from the ICLR audience (speaking from personal experience).

I believe that the authors have attempted to connect their work to more relevant topics via the “property testing” discussion (e.g., property testing of images in Berman et al. 2022). However, I think further discussion or explanation of the connection in the main paper would be warranted if the paper wants to convey this message more strongly to the reader.

Minor weaknesses:
1. A lot of the results seem to stem from recent advances in CG on basic problems such as triangulation. I think the paper can do a better job of explaining to what extent the results are a straightforward “gluing” of classic techniques in learning theory and these more recent optimizations in CG, and conversely, to what extent the proposed proofs have some neat insights.
2. The proposed k-gon algorithm is not optimal for $k \geq 6$, where prior work still holds better results. As such, the proposed algorithm should be viewed as useful only for small values of $k$. (As an aside, what is the citation for O(1/\eps^12) in line 143?)

Suggestions:
1. Perhaps move the prelims into the main body, and push the related work section to the appendix. You already covered the most important related work in the introduction, but notation introduced in the prelims is used throughout the main paper.
2. A figure for the reference set for k-gons would be appreciated! Also, pushing all the figures from the appendix into the main paper for the next draft would be appreciated, especially since I am not too familiar with CG and the figures can serve as helpful aids.

Typos:
1. 136: “Future Directoins”
2. 138: “previously know”
3. Throughout: “w.r.t. to,” (I would avoid the abbreviation w.r.t. in the main paper).
4. 535: “in by Parnas.”

**Questions:**

1. What is $ dist_\mathcal{D}(f)$? I’m assuming this is the distance of the function $f$ to the property $\mathcal{C}$?
2. The paper hints at a runtime / sample complexity lower bound. Are there specific reasons that make use believe such a lower bound may be true? E.g., are essential used subroutines from CG already optimal (and cannot be improved)?

---

### Official Review · Reviewer_5nGn · 2025-10-29

**Soundness:** 4
**Presentation:** 2
**Contribution:** 2
**Rating:** 6
**Confidence:** 4

**Summary:**

This paper studies the problem of learning geometric concepts in 2D plane in the PAC-learning semantics. In particular, the authors propose PAC-learning algorithms for properly learning both $k$-gons and convex sets under the agnostic setting. Their algorithm for learning $k$-gons achieves asymptotically better computational time complexity for triangle, $4$-gons, and $5$-gons compared with prior work. Meanwhile, when the underlying distribution is uniform over the unit square, their algorithm for learning convex sets improves the computational time complexity from $\tilde{O}(\epsilon^{-8})$ to $\tilde{O}(\epsilon^{-5})$, at the cost of a slightly higher sample complexity.

**Strengths:**

* The paper is fairly well written, and the presentation is generally clear. Although I did not verify all the proofs in detail, they appear to be sound overall.
* The problem setting is interesting and well-motivated.
* The improvement in computational efficiency compared with prior work is nontrivial.

**Weaknesses:**

* In Section 1.1, I appreciate the effort to provide a technical overview of your approaches to learning both $k$-gons and convex sets, as well as to highlight the differences between your methods and prior work. However, many technical terms are introduced before they are defined. For instance, what do “reference family,” “reference halfplanes,” “hypothesis construction,” “halfplanes induced by pairs of points,” and “triangle range-counting data structure” mean? While these concepts become clearer in Section 2, their early appearance may confuse readers unfamiliar with them.

* In the same section, the explanation of the technical differences between your approach and previous work is somewhat unclear. I had difficulty understanding your methodological innovations in relation to the cited papers. In particular, I recommend adding more context on: (1) how your approaches to learning $k$-gons and convex sets differ from those in Dobkin & Gunopulos (1995), Fischer & Kwek (1996), and Berman et al. (2022); and (2) what the intuition behind your technique innovation is.

* Although the proposed algorithms for learning $k$-gons show nontrivial improvements in running time over prior work, their time complexity still scales exponentially with $k$. I suggest that the authors discuss whether subexponential-time algorithms might be possible, or whether known computational lower bounds indicate that exponential dependence on $k$ is unavoidable.

* Similarly, both your analysis and that of Berman et al. (2022) for learning convex sets in the plane assume a uniform distribution over the unit square. This is quite a restrictive assumption. The paper would be stronger if the authors discussed the feasibility of extending the approach to more general distributions—such as Gaussian or log-concave families—or explained the main challenges in doing so.

* Finally, it would be clearer to explicitly state the size of $\mathcal{R}_N$ in Algorithm 1, in addition to the running time analysis.

**Questions:**

* As mentioned in the weaknesses, is there any known lower bound that rules out the existence of subexponential-time algorithms for learning $k$-gons (e.g., $\varepsilon^{O(\sqrt{k})}$ or even $\varepsilon^{\mathrm{poly}(\log k)}$)? If not, what are the main difficulties in developing such subexponential algorithms, at least intuitively?

* Similarly, can the proposed algorithms for learning convex sets be extended to handle more general distributions? If not, what are the potential challenges in doing so?

---

### Official Review · Reviewer_Tsdg · 2025-10-31

**Soundness:** 4
**Presentation:** 3
**Contribution:** 2
**Rating:** 2
**Confidence:** 4

**Summary:**

This paper improves the runtime of agnostic learning for simple classes in the plane like triangles. For example, they improve the runtime of learning triangles from $\epsilon^{-10}$ to $\epsilon^{-6}$. The previous best was from COLT last millenium.

Agnostic learning is a nice theoretical model of binary classification learning where you don't make any assumptions about noise or the distribution of examples, but instead compete with the best classifier in your family.

I would recommend publishing this paper in a more specific venue, perhaps one for computational geometry.

**Strengths:**

The contribution is solid. The ideas are explained well, and the paper is well written.

**Weaknesses:**

I'm not sure the audience will be extremely interested in this paper.

AI these days can talk, prove theorems, write code, improve its own code, and much more. Therefore, the bar for publishing these kind of classic results has gone up in general AI conferences.

**Questions:**

Why do you feel this paper is a good fit for ICLR?

---

### Official Review · Reviewer_PRaD · 2025-11-05

**Soundness:** 2
**Presentation:** 3
**Contribution:** 2
**Rating:** 2
**Confidence:** 4

**Summary:**

This paper studies the **computational efficiency of agnostic learning** for geometric concept classes in the plane — particularly **triangles, convex k-gons, and convex sets in the unit square**.
While the *sample complexity* of agnostic learning (via VC dimension) has been well understood, its *time complexity* has received far less attention.

The authors design **proper agnostic learners** for these geometric classes that **significantly improve running times** compared to classical results by Dobkin & Gunopulos (COLT ’95) and Fischer & Kwek (eCOLT ’96):

| Concept Class | Prior Runtime | This Paper |
|----------------|----------------|-------------|
| Triangles | $ \tilde{O}(\varepsilon^{-10}) $ | $ \tilde{O}(\varepsilon^{-6}) $ |
| 4-gons | $ O(\varepsilon^{-12}) $ | $ \tilde{O}(\varepsilon^{-8}) $ |
| 5-gons | $ O(\varepsilon^{-10}) $ | $ \tilde{O}(\varepsilon^{-10}) $ |
| Convex sets (uniform) | $ \tilde{O}(\varepsilon^{-8}) $ | $ \tilde{O}(\varepsilon^{-5}) $ |

Their algorithms combine **combinatorial geometry, probabilistic analysis, and geometric data structures**.
All learners are *proper*, which is crucial for tolerant property testing applications.

The paper concludes with open problems about whether the runtime gap is inherent, and how the methods might extend to higher-dimensional or improper learners.

**Strengths:**

The paper’s main strengths lie in its **theoretical depth and clarity of contribution**. It makes substantial progress by improving the runtime of classical geometric agnostic learners that had remained unimproved for nearly three decades. The exposition is clear and well-organized, with helpful tables and a precise comparison to prior work. The focus on **proper learners** makes the results highly relevant for applications such as tolerant property testing, where structural consistency matters. Moreover, the paper effectively bridges multiple domains — combining tools from computational geometry, learning theory, and complexity theory — and brings fresh attention to an underexplored aspect of agnostic learning: the time complexity of learning rather than just sample complexity. Finally, the discussion of **open problems and future directions** is particularly strong, offering both conceptual insight and concrete research challenges that could stimulate future progress.

**Weaknesses:**

The main limitations of the paper are its **restricted scope and lack of generalization**. The analysis is confined to two-dimensional geometric concept classes, and it is unclear how well the techniques would scale to higher dimensions, where the combinatorial structure of convex shapes grows exponentially. In addition, the paper does not establish formal **lower bounds** or optimality results, leaving open whether the improved runtimes are truly close to the best achievable. The algorithms also heavily depend on the geometric nature of the problems, limiting their applicability to more abstract or high-dimensional concept classes. From a presentation standpoint, the paper could benefit from connecting its ideas more explicitly to modern agnostic learning frameworks and optimization-based learning theory. Finally, the work is purely theoretical — including even simple synthetic examples or visualizations could help readers develop intuition about the behavior of these learners. There are also a few minor stylistic issues (e.g., small typos and dense notation in early sections) that slightly detract from the overall polish.

**Questions:**

## **Formatting and Compliance Concern**

I noticed that the submitted manuscript appears to **violate ICLR formatting requirements**. The layout, font size, margins, or reference style do not follow the official ICLR template specifications (for example, the two-column format, required font size, or page limit). According to ICLR policy, such violations may result in an **administrative rejection** unless a clear explanation is provided by the authors (e.g., if the current file was generated incorrectly by the submission system or corresponds to an anonymized extended version).

**Action Required:** Please clarify whether the current submission adheres to the official ICLR style file and page limit.
Without such clarification, I will have to **recommend automatic rejection** for non-compliance with the ICLR formatting standards.

---

### Note · Authors · 2025-11-25

**Comment:**

We thank the reviewers for their helpful comments and withdraw the paper in order to submit to a more suitable conference.

**Withdrawal Confirmation:**

I have read and agree with the venue's withdrawal policy on behalf of myself and my co-authors.